# Machine-Learning-Based Fatigue Trend Analysis on IMU Wearable Sensor Data from Construction Site Workers

**DOI:** 10.3390/s25247455

**Published:** 2025-12-08

**Authors:** Janne S. Keränen, Jamil Ahmad, Sergio Leggieri, Satu-Marja Mäkelä, Darwin G. Caldwell, Christian Di Natali, Atte Kinnula, Pekka Siirtola

**Affiliations:** 1VTT Technical Research Centre of Finland Ltd., Kaitoväylä 1, P.O. Box 1100, 90571 Oulu, Finland; satu-marja.makela@vtt.fi (S.-M.M.); atte.kinnula@vtt.fi (A.K.); 2Department of Advanced Robotics (ADVR), Istituto Italiano di Tecnologia, via Morego, 30, 16163 Genoa, Italy; jamil.ahmad@iit.it (J.A.); sergio.leggieri@iit.it (S.L.); darwin.caldwell@iit.it (D.G.C.); christian.dinatali@iit.it (C.D.N.); 3Department of Informatics, Bioengineering, Robotics and Systems Engineering (DIBRIS), Università degli Studi di Genova (UniGe), 16145 Genova, Italy; 4Biomimetics and Intelligent Systems Group, Faculty of Information Technology and Electrical Engineering, University of Oulu, Pentti Kaiteran katu 1, P.O. Box 8000, 90014 Oulu, Finland; pekka.siirtola@oulu.fi

**Keywords:** fatigue, wearables, sensor, IMU, machine learning

## Abstract

Physical fatigue is a major cause of work-related accidents and musculoskeletal injuries in the construction industry, and additional means are needed for their identification and management to prevent long-term consequences. Based on recent scientific literature, fatigue can be detected with wearable inertial measurement units (IMUs). However, IMUs for detecting fatigue have been so far tested mainly in the laboratory; therefore, a research gap exists in application of IMU sensors for detecting fatigue in real-life work settings. The aim of this paper is to bring the fatigue trend detection with IMUs closer to real-life context by using wearable IMU sensor data from an actual construction site measuring actual workers with simulated work tasks. The paper also presents advancements in fatigue trend detection with frequency domain investigations to gain access to more detailed fatigue relevant features. Machine-learning methods are used to predict fatigue trends based on IMU data, resulting in fatigue trend detection accuracy that advances the state of the art. More knowledge is also unearthed about relevant sensor locations and features.

## 1. Introduction

In the US construction industry, 33% of all work-related musculoskeletal injuries are attributed to overexertion [1]. In addition, physical fatigue is found to be a major cause of work-related accidents in the building construction industry [2,3]. Therefore, detecting and tracking fatigue is important, so that timely interventions (e.g., breaks) can be introduced [4]. To support physical ergonomics and overall safety at work, personalized fatigue assessments would be beneficial.

Fatigue is usually detected by measuring heart rate (HR) and heart rate variability (HRV), skin temperature, muscle jerk (IMU) and muscle response (electromyography, EMG) metrics [5]. However, HR, HRV, skin temperature and EMG all require a sensor with skin contact, which can be considered obtrusive. Fatigue adversely affects movement control and quality, therefore, demonstrating increases in the jerk values (with IMUs) [6]. In [7], features associated with jerk and acceleration at the wrists and hips were better predictors of physical fatigue than features associated with heart rate [7]. Heart rate data (HRV) might still be applicable to evaluate general fatigue; however, jerk metrics might be more suitable to assess localized fatigue (e.g., lower back, lower limbs) [5,7,8]. Then again, heart rate might not be directly proportional to the individuals’ fatigue state, but rather an indicator of the intensity of the current task [9]. Jerk metrics, on the other hand, can be sensitive to sensor position and the current task (task- and person-specific cutoff values) [5].

Based on the scientific literature presented in the previous paragraph, fatigue can be detected with wearable sensor IMUs. IMUs also present other benefits: IMU sensors are cheap, they can be easily integrated into clothing, and they are relatively unobtrusive. Another clear advantage of IMUs is that the data that they produce is less privacy-intrusive than those captured by, e.g., cameras or health-related sensors, such as heart rate monitors. However, IMUs for detecting fatigue have so far been mainly tested in the laboratory [5]; therefore, a research gap exists in the application of IMU sensors for detecting fatigue in real-life settings, where the volatile environment presents challenges. This research will aim to cover this gap and contribute to the field of wearables and personal wellbeing.

The inspiration for the research presented in this paper is based on previous work [10], where the aim was to detect hazardous situations and accidents by applying human-activity recognition methods on wearable sensor data. A dataset [11] was collected with inertial motion unit (IMU) sensors from real construction work conditions and was analyzed using machine-learning models to recognize physical activities of the workers. The project revealed challenges in both reliable data collection (sensor placements, battery consumption, sensor synchronization, etc.), and the challenge of tracking data from workers, which would reveal essential parts of their wellbeing and safety. However, initial assessments of the sensor data in the frequency domain to track overall fatigue also raised the interest in the possibility of a more detailed setup for tracking fatigue itself.

The research presented in this paper aims to answer the question: How can the fatigue trends of workers in real work environments be accurately detected with non-intrusive IMUs? The methods include algorithms such as pre-processing, filtering, and other frequency domain manipulations to gain access to fatigue relevant features. For the actual data analysis, machine-learning processes and methods were used to detect and classify fatigue and activities. One hypothesis for a potential metric for fatigue build-up over time could be the change related to a starting “normal” baseline. Although fatigue is presumably person-dependent, a general model approach was investigated to find a generalized fatigue model for multiple people. The efficacy of the fatigue models used was benchmarked against existing results from state-of-the-art literature. All these algorithms were used with workers’ data in a real-life construction scenario. During the test, the workers wore the Smart Suit developed at the Istituto Italiano di Tecnologia [12].

The work presented in this paper has been funded by BEEYONDERS EU project. BEEYONDERS’s problem statement is highly relevant regarding the aims of this paper too. Demographic trends in Europe are posing a challenge, e.g., to the construction sector, which is facing labor shortage, and an aging existing workforce. This can be addressed through the integrated adoption of breakthrough technologies like smart wearables and artificial intelligence, which can help construction tasks to be completed with more efficacy, while at the same time, making the sector more attractive to young people, and improving the safety, productivity, quality, and environmental impact of construction projects. The project aims to demonstrate the safety of wearables in a construction environment in cooperation with workers and increase the wellbeing of the construction workforce concerned and develop solutions for monitoring the wellbeing of the workforce and prevention of accidents [13].

In this paper, we present novel contributions: the use of frequency plane (PSD) peaks as features in fatigue detection methodology; these peaks represent the major frequencies of the human movement and illustrate shift in the intensity of the movement. Our proposed PSD features are strongly present in the most important features listing, hinting at their high usability in detecting fatigue trends compared to jerk values and standard machine-learning features. Our approach also achieved promising MAE rates, improving the state of the art regarding the accuracy of IMU-based detection of construction site fatigue trends. We also emphasize that another strength of this research is that the Technology Readiness Level (TRL) of the dataset and approach used in this paper is relatively high, TRL 6, since it is from the actual construction site with actual construction workers.

## 2. Dataset Description and Machine-Learning Methodology

This chapter aims to describe method development for fatigue trend analysis with wearables for construction workers. The ideal scenario would be to use a minimum amount of non-intrusive IMU sensors to enable realistic usage at work sites; however, during initial method development, all possible sensors and features were used in assessing the fatigue trend detection accuracy.

### 2.1. Smart Suit and Data Acquisition

The fatigue trend detection uses data coming from the Smart Suit. The Smart Suit has been developed at Istituto Italiano di Tecnologia (IIT) to analyze and monitor worker movements and muscle activations in operational scenarios. Several sensors, e.g., IMU and EMG, have been integrated within a high-visibility Personal Protective Equipment (PPE) in a seamless way, avoiding bulky external devices on the worker. Smart Suit provides a wearable motion capture technology with improved usability for the final user. A patented mechanism ensures the sensor’s anchoring and stability during usage.

The Smart Suit integrates, under its outmost layer, ten magnetometer-free IMUs, making this technology particularly suitable for industrial applications [14]. More details are reported in [12]. Small portable batteries power the sensors and the Raspberry Pi (RPI), the single-board computer. The RPI equips an external data storage device with a capacity of 16 GB that allows continuous data logging of more than two hundred variables at 200 Hz for about 4 h.

The Smart Suit software architecture consists of a distributed framework of nodes on multiple levels. The Core node is responsible for loading, initializing, and managing low- and high-level nodes. The Core node also ensures the synchronization of all the other nodes, such as the human activity recognition [15] and the ergonomic risk assessment [16]. The Streamer node can be used as a bridge between the Smart Suit data and variables, and external IoT devices among the high-level modules.

The data used in this research is measured from six positions of the Smart Suit IMU sensors: left leg, right leg, left arm, right arm, sternum, and pelvis. Arm data are important considering activities that are performed mainly by hand or with the upper body; in these cases, the detection of fatigue could be challenging without the arm data. The Smart Suit was used by IIT to collect data from a field study at a construction site located in Helsinki, Finland. The field study was performed in a dedicated real-life production environment with 5 individual workers (age 30 ± 17 years, height 176 ± 6 cm, and weight 80 ± 4 kg) performing work activities on five different days. The experiment was approved by the Ethical Committee of Liguria (protocol reference number: CER Liguria 001/2019) and complies with the Helsinki Declaration. All the subjects signed an informed consent form prior to participating, after receiving a full explanation of the experimental procedure.

One of the activities that the Smart Suit was tested on workers was hauling. This activity consisted of lifting a heavy object, carrying it a few meters and lowering it, and repeating this cycle. The task was performed between two loading stations placed approximately 10 to 12 m apart from each other. The workers carried out three consecutive tasks, first carrying bricks (total weight 10 kg) 4–5 times, then carrying a cement bag (25 kg), and after this, executing a free walk (no load). This sequence of movements was repeated for the whole duration of the test, and the sequence aimed at keeping the strenuousness of the lifting in a safe range. The duration of every activity was individual to the worker, and the test lasted between 30 and 60 min, depending on the test subject’s pace. The test was performed on two separate days; altogether, two repetitions were recorded for each worker and each task, apart from worker 4, who did only one repetition on day 4, and worker 3, who did not perform on day 2. This resulted in 8 datasets for hauling activities. Table 1 illustrates the overall view for the hauling task datasets.

Along with the Smart Suit data, self-reported effort assessments (force exerted during an activity) were collected from the workers in the beginning, in the middle and at the end of the work tasks. The effort assessment was on a scale of five values, from 1 to 5. Table 2 details the effort levels.

### 2.2. Pre-Processing

Only the accelerometer data coming from the Smart Suit IMUs went through pre-processing. There was some slight variance in the data frequency and occasional NaN values in the timestamps, but the approximate average frequency of the data was 45 Hz. The accelerometer time series data was pre-processed by checking the average frequency of each sensor, and calculating the total acceleration from the x, y, and z axes of the accelerometer data. Possible null values were filtered out. A low-pass Butterworth filter with a cutoff frequency of 18 Hz was added for the total acceleration data. The accelerometer data was windowed with sliding window technique, with 12 min-long windows sliding 1 min at a time.

The effort values (force exerted during activity) were pre-processed so that the values attained for each worker during a task (from the beginning, from the middle and from the end), are linearly interpolated so that they span the whole task time, providing a target value for each of the time windows generated from the data for the task. These interpolated effort values operate as target values (a baseline) that are used in guiding the machine-learning training and testing.

### 2.3. Features

The feature calculation phase utilized power spectral density-based features that were already used in the previous project [11]. The power spectral density (PSD) was calculated using Welch’s method [17], using fast Fourier transform to reveal the frequency plane details of the movement data from the accelerometers. From the PSD, the predominant frequencies were analyzed for each time window, and especially focused on the predominant frequency peaks, and their change over time. These peaks are illustrated in Figure 1. The frequency values of the 8 highest peaks of each PSD time window were used as features. In addition, a power weighed sum of these frequency values was calculated and used as a feature.

The power weighed sum of the frequency values is calculated as follows:(1)$$x=\sum_{k=1}^{8}f_{k}{\;p}_{k},$$
where *x* is the power weighed sum, *f* is the frequency of each frequency peak, and *p* is the power of each frequency peak.

Other features that were calculated can be divided into standard features and jerk features. Standard features are common features used in machine learning, including mean, standard deviation, median, 75th percentile, and 25th percentile. These features are calculated directly for the windowed total acceleration values. Jerk features are otherwise the same, except the total acceleration windowsare first used to calculate the jerk value, i.e., derivative/gradient of the aforementioned time window. These jerk time windows are then used to calculate the machine-learning features of mean, standard deviation, median, 75th percentile, and 25th percentile. Calculating all these features for all the 6 different sensors resulted in a total of 114 different features being used in the feature selection phase. After feature calculations, the features were normalized by dividing each feature value with the average of the first two values of each feature vector. The normalization is needed to create features that are comparable between different test subjects and to be able to create a generalized fatigue model from many sources. Different data standardization approaches were tried, such as setting the dataset mean to zero and standard deviation to 1 (standard scaling), and by scaling each feature to a given value range (from 0 to 1), but the normalization with the equation below yielded the best results.(2)$$f_{normalized}=f_{x}\;/\;(f_{1}+{\;f}_{2})$$

For splitting the data into train and test splits, the leave-one-out method was used, where each of the datasets is left as a test case in their turn and the other datasets are used for model training., e.g., in the case of the hauling task, there are 8 different permutations of the train/test split.

For feature selection, univariate feature selection was used, in which univariate statistics are used to determine the best features. More specifically, variance F-value was analyzed for the feature samples [18]. This is more commonly used in classification tasks but also works well for regression. The feature selection results in the five best features chosen for each train/test split for model training. A random forest regressor was used in model training with the following parameters: the number of trees in the forest is 80; for the function to measure the quality of a split, the mean squared error was used with Friedman’s improvement score, and randomness of the bootstrapping of the samples was used when building trees were set to an integer value of 0 [19]. The whole algorithmic process is explained in Section 2.2 and Section 2.3 and is illustrated in a flowchart in Figure 2.

## 3. Fatigue Trend Analysis Results

To evaluate the accuracy of the trained fatigue models, the mean absolute error (MAE) was used, which is one of the most appropriate ones to assess how far a predicted fatigue trend curve is from the actual one. MAEs were calculated for each of the train/test splits (for all the datasets), and an overall value was obtained by calculating the mean value of all the calculated MAEs. These values are presented in Table 3. The overall average MAE for all workers is 0.3670.

Our study also provides results regarding the most important sensors and features while testing the fatigue model. Each train/test split dataset with its five best features produced feature importance results. The importance of a feature is computed as the (normalized) total reduction of the criterion brought by that feature, which is also known as the “Gini importance” [19]. Sum scores for the best features of all eight datasets were calculated, and they are depicted in Table 4.

From the information in Table 4, we can see that the pelvis and sternum areas produce the most meaningful features regarding fatigue accumulation. This also became evident when additional model training was performed using only the features from pelvis and sternum sensors, resulting in an average MAE of 0.419, and further on, when only using features from the pelvis sensor, the average MAE was 0.47, resulting in a relatively small reduction in accuracy (0.3670 when using all sensors) even though data from majority of the sensors was not used. Also, interestingly, our proposed new PSD features are strongly present in the most important features, hinting their high usability in detecting fatigue trends compared to jerk values and standard machine-learning features. With additional model training without using the PSD related features, the average MAE worsens to 0.533. Figure 3 depicts all the effort trends predicted by the fatigue model (using all possible features), and the linearly interpolated actual expected values. The fatigue trend detection is well in line with the expected effort values (effort self-assessment collected from workers).

Regarding the model used, random forest regression was chosen after trials with other regression models. In trials, the same model parameters were used for other models as for random forest regression (where applicable); otherwise, default parameters were used. Regression itself was chosen as fatigue over time is, by nature, a continuous and cumulative trend depending on many input variables. Model trial results are depicted in Table 5.

## 4. Discussion

The research presented aims to answer the question: How can the fatigue trends of workers in real work environments be accurately detected with non-intrusive IMUs? The data from the construction site provided a solid foundation for the machine-learning work on IMU-based fatigue trend detection. Starting with the pre-processing, we chose the initial value-based normalization based on experiences presented in [20] and modified it to better represent the initial non-fatigue state. We tried also other data standardization approaches, but the normalization yielded best results. We also used Butterworth low-pass filtering as in [20] and found that a higher cutoff value of 18 Hz (often used around 10 Hz) provided more detailed features, especially considering our use of PSD features that rely on the frequency plane data. The data was windowed with the sliding window technique, with 12 min-long windows sliding 1 min at a time. Through experimentation, this was found to be the most accurate time-windowing approach compared to shorter time windows, or non-overlapping time-windowing methods.

The literature suggested that supervised machine-learning methods such as random forest [20,21] and regression models [7] are good starting points. Our experiments proved that random forest regression was the best model for fatigue trend detection purposes. Other models were tried, but with more inaccurate results, as depicted in Table 5.

The usage of jerk features is a common phenomenon in the fatigue detection literature like in [7,21], and our novel contribution is the use of frequency plane (PSD) peaks that represent the major frequencies of the movement. PSDs have been previously used in fatigue estimation as in [22,23], but to our knowledge, the PSD peaks have not been utilized in fatigue estimation before. Our proposed PSD peak-based features are strongly present in our most important feature listing, hinting at their high usability in detecting fatigue trends compared to jerk values and standard machine-learning features. Pelvis and sternum areas produced the most meaningful features regarding fatigue accumulation.

Cross-validation is a common method in machine learning, and in the IMU-based fatigue trend detection literature. Compared to work in [7], the strength of our fatigue model method is that all the datasets can be used in testing in cross-validation, not just choosing a couple of the datasets as the testing datasets. All the effort trends predicted by the fatigue model were well in line with the expected effort values and the overall average accuracy (MAE) for all workers is 0.3670. The authors in [7] achieved a MAE value of 2.16 for their fatigue trends. To be able to fully compare to this, we need to calculate the percentage of the MAE in the context of the used effort scale. The authors of [7] used a scale from 6 to 20 with 15 steps; thus, their percentage error was 2.16/15 = 0.144, resulting in a 14.4% MAE in the context of the used scale. Our effort scale was from 1 to 5 with 5 steps, and our percentage error was 0.3670/5 = 0.0734, resulting in a 7.34% MAE in the context of the scale used, thus markedly improving the state of the art regarding the accuracy of IMU-based detection of construction site fatigue trends.

Another fatigue trend detection example from the related field of manufacturing is where the authors [24] used a wearable system, including electrocardiogram (ECG), heart rate (HR), skin temperature and motion signals from the accelerometers and gyroscopes. Test subjects carried out simulated manufacturing tasks with a weighted vest to exaggerate the induced fatigue. The best fatigue trend MAE the authors achieved was 2.01 on a fatigue scale from 0 to 7 with 8 steps; thus, their percentage error was 2.01/8 = 0.25125, resulting in 25.13% [24].

When considering other fatigue trend detection methodologies, an approach [25] was presented by authors by using an electroencephalography (EEG) sensor to monitor brain activity and a photoplethysmography (PPG) sensor to monitor cardiovascular parameters. The authors introduced laboratory-induced fatigue with experimental procedures including cycling and jumping rope. Their ensemble model reached fatigue trend detection with average mean squared error (MSE) values of 0.0007 [25].

For comparison purposes, we calculated MSEs for our datasets between effort predictions by our model and the expected effort values, resulting in our approach having an average MSE of 0.25. This demonstrates that there is a considerable gap in accuracy between gold-standard sensors used in a laboratory by the authors of [25], and more practical sensors used in actual construction sites.

On the other hand, the authors in [26] presented an approach, where the research monitored fourteen construction workers’ fatigue in an on-site work on the repair and maintenance of a utility tunnel and gathered physiological measures and fatigue data simultaneously. Multiple sensors, including electrodermal activity (EDA), skin temperature (ST), and photoplethysmography (PPG), which can monitor heart rate (HR) and heart rate variability (HRV) were used (as opposed to our IMU-only approach). Correlation analysis between ratings of fatigue (ROF) and each physiological measure was performed, resulting in a best correlation coefficient value of 0.33 for EDA [26].

For comparison, we calculated correlation coefficients for our datasets between effort predictions by our model and the expected effort values, resulting in an average correlation coefficient of 0.38. Our approach was deemed to have slightly better accuracy results, but our work tasks were simulated, while the authors in [26] used more intrusive sensors, but tracked actual real-life work tasks. This demonstrates the special challenge of tracking real-world construction site fatigue, since the work tasks at the construction site are multifaceted and vary in pace. Some tasks might induce major fatigue (e.g., heavy hauling), while others do not necessarily manifest in physical fatigue (e.g., electric wiring, door lock installations, etc.) This challenge of gathering fatigue data from real-life environments also manifests itself in our approach, since fatigue levels 4 and 5 were not reported by the workers.

Besides presenting a novel approach that improves the state of the art regarding construction context, we would also emphasize that the strength of this research is that the TRL of the dataset and approach used in this paper is relatively high, around TRL 6, since it is from the actual construction site and from actual construction workers. For comparison, the authors in [7,20,21,25] use laboratory data, which is usually around TRL 4. The exception is [26], where the authors reached TRL 7.

Interesting and most certainly challenging future research would be on the utilization of IMUs and fatigue models in actual everyday construction site work in long-term tests, spanning several weeks at minimum. Organizing this kind of data gathering is highly challenging but would also provide valuable information regarding the most informative sensor locations and features in combination with the activities being performed. Such a dataset would also enable the exploration of personalized fatigue models, and their performance against a general fatigue model. Initially, we intended to use personalized fatigue models to gain more articulate fatigue modeling, but the results were not encouraging, partly due to the low number of different datasets from the same people. A more general fatigue model, where mixed datasets from different people are used for training (besides the test dataset in question during cross-validation) yielded the most encouraging results. In this more general approach, most people had data in training and test phases, besides workers 3 and 4, who had only one dataset and thus had no data in training during their testing phase. Interestingly, the MAEs of workers 3 and 4 were among the lowest, besides having no data in the training phase. Another interesting future work aspect would be the comparison of the hauling task to another kind of work to see possible differences in IMU-based fatigue trend detection accuracies. Although this research focuses on the construction industry, other industries such as steelworks, forestry and healthcare also face the same issues of physically demanding work that could benefit from the fatigue analysis.

## 5. Conclusions

Physical fatigue is found to be a major cause of work-related accidents and injuries in the construction industry, and additional means are needed for their identification and management. Fatigue can be detected with wearable IMUs, but a research gap exists in application of IMU sensors for detecting fatigue in real-life work settings. This research aims to bring the fatigue trend detection with IMUs closer to the real-life context by using a dataset with a TRL of 6, since it is from an actual construction site with actual construction workers.

We presented a novel contribution: the use of frequency plane (PSD) peaks as features in fatigue detection methodology. These peaks represent the major frequencies of the human movement and illustrate shift in the intensity of the movement. Our proposed PSD features are strongly present in the most important features listing, hinting at their high usability in detecting fatigue trends compared to jerk values and standard machine-learning features. Our approach also achieved promising MAE rates, markedly improving the state of the art regarding the accuracy of IMU-based construction site fatigue trend detection. More knowledge is also unearthed about the most relevant sensor locations and their machine-learning features.

## Figures and Tables

**Figure 1 sensors-25-07455-f001:**
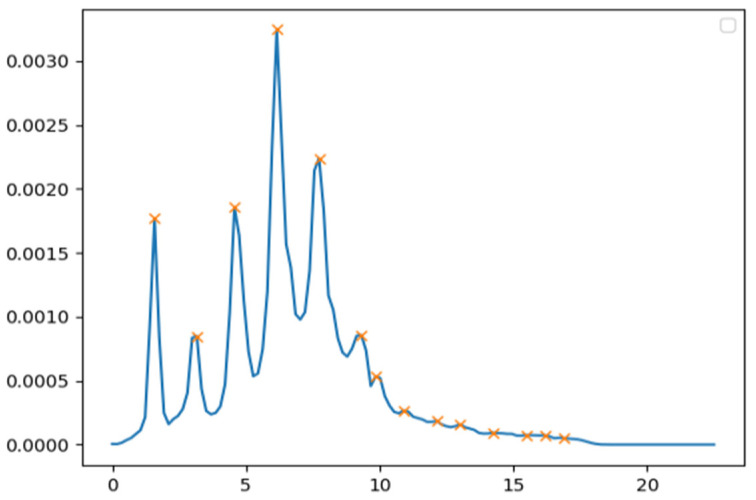
Frequency peaks of power spectral density of the one time-window of real worker IMU data.

**Figure 2 sensors-25-07455-f002:**
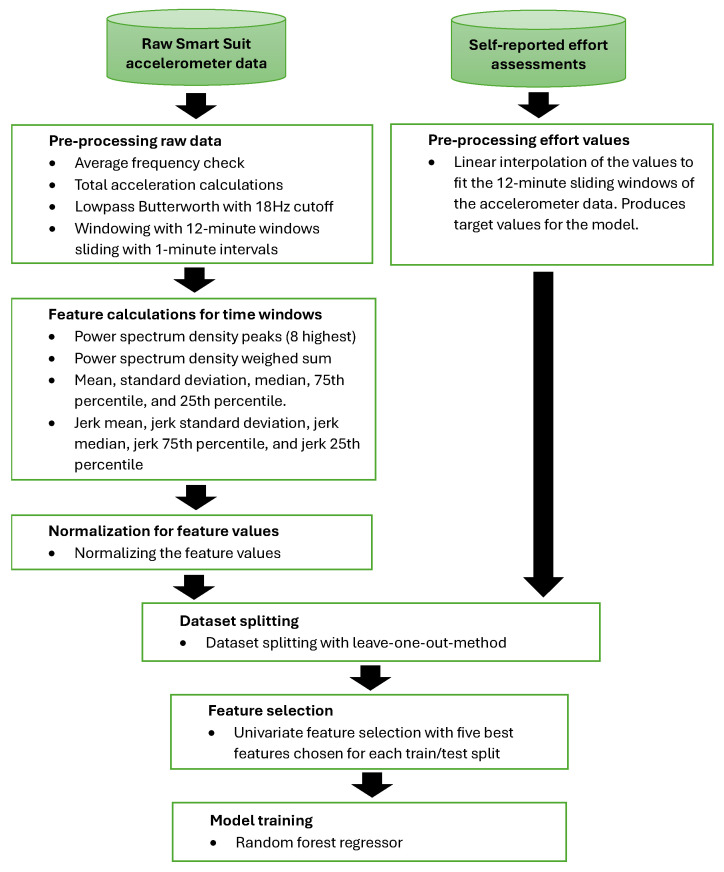
Flowchart depicting the algorithmic process.

**Figure 3 sensors-25-07455-f003:**
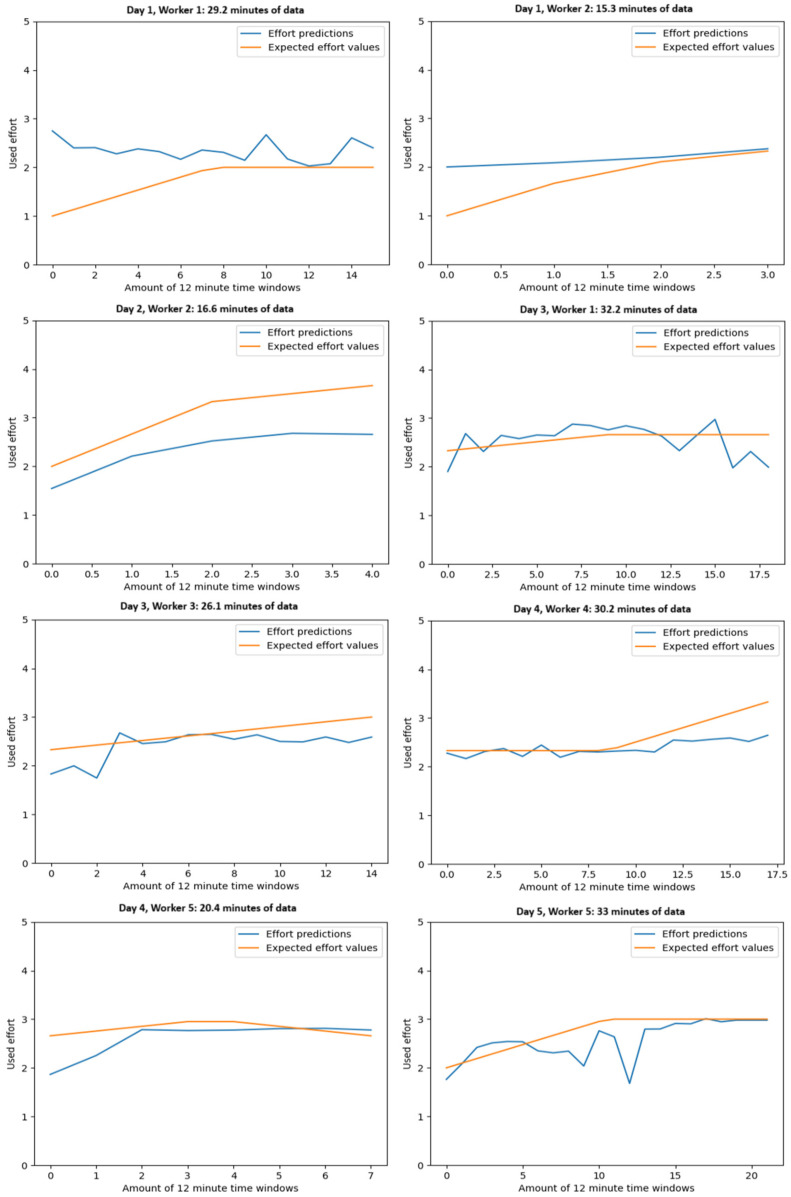
Resulting effort predictions and expected values for all hauling datasets.

**Table 1 sensors-25-07455-t001:** Overall view on the hauling task datasets.

Day	Worker	Dataset Duration
1	1	29.2 min
1	2	15.3 min
2	2	16.6 min
3	1	32.2 min
3	3	26.1 min
4	4	30.2 min
4	5	20.4 min
5	5	33.0 min

**Table 2 sensors-25-07455-t002:** Used effort values and their explanations.

Force Exerted During Activity	Explanation
1	Low
2	Normal
3	Moderate
4	High
5	Very high

**Table 3 sensors-25-07455-t003:** Accuracies for all the datasets.

Day	Worker	Data Split MAE
1	1	0.6082
1	2	0.3915
2	2	0.7070
3	1	0.2441
3	3	0.2724
4	4	0.2264
4	5	0.2432
5	5	0.2428
Average MAE for all workers		0.3670

**Table 4 sensors-25-07455-t004:** The most important sensors and features.

Sensor Location and Feature	Sum Feature Importance
Pelvis median	2.452521
Sternum PSD power weighed sum	1.900824
Pelvis 8. PSD peak	1.335845
Right upper arm median	0.786098
Right upper arm 4. PSD peak	0.501262
Pelvis jerk standard deviation	0.354390
Right thigh jerk mean	0.154660
Right upper arm 75 percentile	0.149535
Right upper arm 2. PSD peak	0.138543
Right thigh 7. PSD peak	0.136377
Left upper arm 4. PSD peak	0.083994
Left thigh 2. PSD peak	0.005953

**Table 5 sensors-25-07455-t005:** Accuracies for the different tried regression models.

Model Tried	Average MAE for all Train/Test Splits
Random forest regression	0.366951
Gradient-boosting regression	0.414846
Decision tree regression	0.432119
Lasso regression	0.471001
Support vector regression	0.478087
Ridge regression	0.561196

## Data Availability

The datasets generated during the current study are available from IIT: christian.dinatali@iit.it on reasonable request.

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
