# Peer review of "Machine-Learning-Based Fatigue Trend Analysis on IMU Wearable Sensor Data from Construction Site Workers"

_sensors, 2025, doi:10.3390/s25247455_

Round 1
Reviewer 1 Report
Comments and Suggestions for Authors
The authors state that “IMUs for detecting fatigue have been so far tested mainly in laboratory, therefore a research gap exists in application of IMU sensors for detecting fatigue in real-like work settings.” Further they state: “This research will aim to cover this gap and contribute to the field of wearables and personal wellbeing.”
My comment explains that it is not only the construction industry where safety at work is an important issue. For example, current research in the forestry sector is reflecting the situation of physically demanding professions, manual work and specific outdoor conditions. Fatigue of forestry workers, among other topics, is under study and not only in laboratory setting. Moreover, the IMUs are used also, which could be an inspiration for the authors of this study. This is the reason why I recommend for example this study:
Röhrich, M., Abramuszkinová Pavlíková, E., & Šácha, J. (2025). Biomechanical Analysis of Gait in Forestry Environments: Implications for Movement Stability and Safety. Forests, 16(6), 996. https://doi.org/10.3390/f16060996
Similarly, the forestry sector could be also an inspiration for the topic of fatigue and the measurement of HR and HRV. For example, the following field-study focused on timber harvesting operations and the study introduced a simplified index methodology for workload assessment. Specifically, Garmin, Biostrap, and Whoop devices, were used and the dependence of the heart rate (HR) on physical workload was examined to calculate the Heart Rate Index. These results can be used in common practice for all workers’ self-assessment to increase safety and health protection at work, also in the construction sector.
The details are mentioned for example in this paper:
Abramuszkinová Pavlíková, Eva, Pavel Nevrkla, and Martin Röhrich. 2024. "Heart Rate Index as a Measure of Physical Workload in Chainsaw Operations" Applied Sciences 14, no. 24: 11483. https://doi.org/10.3390/app142411483
I believe that the inspiration from the implementation of digital technologies in the field research in the forestry sector could be beneficial for setting the text under review in a wider context. I suggest authors consider expanding the introduction section in this respect.
Author Response
Comment 1:
The authors state that “IMUs for detecting fatigue have been so far tested mainly in laboratory, therefore a research gap exists in application of IMU sensors for detecting fatigue in real-like work settings.” Further they state: “This research will aim to cover this gap and contribute to the field of wearables and personal wellbeing.”
My comment explains that it is not only the construction industry where safety at work is an important issue. For example, current research in the forestry sector is reflecting the situation of physically demanding professions, manual work and specific outdoor conditions. Fatigue of forestry workers, among other topics, is under study and not only in laboratory setting. Moreover, the IMUs are used also, which could be an inspiration for the authors of this study. This is the reason why I recommend for example this study:
Röhrich, M., Abramuszkinová Pavlíková, E., & Šácha, J. (2025). Biomechanical Analysis of Gait in Forestry Environments: Implications for Movement Stability and Safety. Forests, 16(6), 996. https://doi.org/10.3390/f16060996
Response 1:
Thank you for the review comments. The reviewer commented that it is not only the construction industry where safety at work is an important issue, which we agree. Besides construction industry and forestry which the reviewer referenced, steelworks, and healthcare face the same issues of physically demanding manual work. The referenced work about biomechanical analysis of gait in forestry is interesting but tangential for our manuscript. We added to the discussion an expansive sentence hinting the applicability of fatigue tracking not only for construction, but also for numerous other fields of manual work. All modifications done should be found with “Track Changes” feature of the Word document.
Comment 2:
Similarly, the forestry sector could be also an inspiration for the topic of fatigue and the measurement of HR and HRV. For example, the following field-study focused on timber harvesting operations and the study introduced a simplified index methodology for workload assessment. Specifically, Garmin, Biostrap, and Whoop devices, were used and the dependence of the heart rate (HR) on physical workload was examined to calculate the Heart Rate Index. These results can be used in common practice for all workers’ self-assessment to increase safety and health protection at work, also in the construction sector.
The details are mentioned for example in this paper:
Abramuszkinová Pavlíková, Eva, Pavel Nevrkla, and Martin Röhrich. 2024. "Heart Rate Index as a Measure of Physical Workload in Chainsaw Operations" Applied Sciences 14, no. 24: 11483. https://doi.org/10.3390/app142411483
I believe that the inspiration from the implementation of digital technologies in the field research in the forestry sector could be beneficial for setting the text under review in a wider context. I suggest authors consider expanding the introduction section in this respect.
Response 2:
The referenced work about HR based index methodology for workload assessment was interesting and could provide inspiration for future work with IMU based fatigue tracking, at least in the form of supporting the baseline studies along with worker questionnaires. Thank you for pointing that out.
Reviewer 2 Report
Comments and Suggestions for Authors
The research is aimed to the problem of the physical fatigue detection using wearable inertial measurement units (IMUs) and to address the point of using actual construction site measurements. Authors proposed thr power spectrum density peak-based features to detect the fatigue trends which is stated as a novel contribution.
Author Response
Comment:
The research is aimed to the problem of the physical fatigue detection using wearable inertial measurement units (IMUs) and to address the point of using actual construction site measurements. Authors proposed thr power spectrum density peak-based features to detect the fatigue trends which is stated as a novel contribution.
Response:
Thank you for the review comments. All modifications done should be found with “Track Changes” feature of the Word document.
Reviewer 3 Report
Comments and Suggestions for Authors
This paper endeavors to fill a research gap in terms of IMU fatigue detection. Nevertheless, the rigor and rationality of this study are open to question. While much effort is needed for improving the manuscript, several comments and suggestions are listed below for reference.
- Page 3, In “Dataset description and machine learning methodology”. This article outlines the entire algorithmic process, but it does not include a flowchart, which could have made it easier for readers to understand the content.
- Page 5, In“3 Features”.The article mentions the feature extraction section. However, feature selection is also crucial. The article needs to explain why these features were chosen.
- Page 6, Table 3 & Page 7, Table 5. There's no need to mention both MAE and accuracy; just specify one metric is enongh. Otherwise, it will cause confusion.
- Page 8, Figure 2. Results are a lbit redundant. Only representative results need to be displayed and analyzed.
Author Response
Comments:
This paper endeavors to fill a research gap in terms of IMU fatigue detection. Nevertheless, the rigor and rationality of this study are open to question. While much effort is needed for improving the manuscript, several comments and suggestions are listed below for reference.
Page 3, In “Dataset description and machine learning methodology”. This article outlines the entire algorithmic process, but it does not include a flowchart, which could have made it easier for readers to understand the content.
Page 5, In“3 Features”.The article mentions the feature extraction section. However, feature selection is also crucial. The article needs to explain why these features were chosen.
Page 6, Table 3 & Page 7, Table 5. There's no need to mention both MAE and accuracy; just specify one metric is enongh. Otherwise, it will cause confusion.
Page 8, Figure 2. Results are a lbit redundant. Only representative results need to be displayed and analyzed.
Response:
Thank you for the review comments. The reviewer points out that a flowchart would be beneficial to explain the algorithm methodology, which we agree, and added into the manuscript (Figure 2). The feature selection process is explained at the end of chapter 2.3 (univariate feature selection with 5 best features chosen for each split) and is also now included in the flowchart. In Tables 3 and 5, and for some of the text, we reduced the usage of “accuracy” and used only the “MAE” to reduce duplicate terminology. We believe Figure 2 gives readers more detailed information about how the predicted and actual fatigue cumulates over time and thus gives more comprehensive information about the performance of the fatigue mode. All modification should be found with “Track Changes” feature of the Word document.
Round 2
Reviewer 3 Report
Comments and Suggestions for Authors
The manuscript has been refined and can be accepted.